# Enhancing the Stability of Medium Range and Misalignment Wireless Power Transfer System by Negative Magnetic Metamaterials

**DOI:** 10.3390/ma13245695

**Published:** 2020-12-14

**Authors:** Songcen Wang, Cheng Jiang, Xiong Tao, Feng Chen, Cancan Rong, Conghui Lu, Yingqin Zeng, Xiaobo Liu, Renze Liu, Bin Wei, Minghai Liu

**Affiliations:** 1China Electric Power Research Institute, Beijing 100192, China; wscen@epri.sgcc.com.cn (S.W.); jiangcheng@epri.sgcc.com.cn (C.J.); weibin@epri.sgcc.com.cn (B.W.); 2State Key Laboratory of Advanced Electromagnetic Engineering and Technology, School of Electrical and Electronic Engineering, Huazhong University of Science and Technology, Wuhan 430074, China; xiong_tao@hust.edu.cn (X.T.); ccrong@hust.edu.cn (C.R.); conghuilu@hust.edu.cn (C.L.); yqzeng@hust.edu.cn (Y.Z.); xbliu@hust.edu.cn (X.L.); liurenzhe@hust.edu.cn (R.L.); 3State Grid Zhejiang Electric Power Co., Ltd., Hangzhou 310007, China; chenfeng@evs.sgcc.com.cn

**Keywords:** negative magnetic metamaterials, lateral misalignment, angular offset, wireless power transfer

## Abstract

The misalignment of the resonant coils in wireless power transfer (WPT) systems causes a sharp decrease in transfer efficiency. This paper presents a method which improves the misalignment tolerance of WPT systems. Based on electromagnetic simulations, the structural unit parameters of the electromagnetic material were extracted, and an experimental prototype of a four-coil WPT system was built. The influence of electromagnetic metamaterials on the WPT system under the conditions of lateral misalignment and angular offset was investigated. Experiments showed that the transfer efficiency of the system could be maintained above 45% when the transfer distance of the WPT system with electromagnetic metamaterials was 1 m and the resonant coils were shifted laterally within one coil diameter. Furthermore, the system transfer efficiency could be stabilized by more than 40% within an angle variation range of 70 degrees. Under the same conditions, the transfer efficiency of a system without electromagnetic metamaterials was as low as 30% when lateral migration occurred, and less than 25% when the angle changed. This comparison shows that the stability of the WPT system loaded with electromagnetic metamaterials was significantly enhanced.

## 1. Introduction

Wireless power transfer (WPT) technology has matured in recent years [1,2,3,4,5,6], and has been widely applied in fields such as electric vehicles, portable medical devices, and mobile phone charging. However, some problems remain to be solved, such as electromagnetic pollution, low transmission efficiency at medium and long distances, poor system stability, and so on. This paper focuses on the research of system stability solutions.

In WPT systems, stability decreases, especially power transfer, when the resonant coil changes, which weakens the practicability of WPT technology. The authors of [7] summarized the effects of the misalignment of the resonant coil, e.g., due to angular offset and lateral misalignment, which reduces the stability of the system. In recent years, research on WPT has put forward many schemes to improve system stability. In [8], flux pipe couplers were added between transfer channels to improve system tolerance to coil offset. By comparing a multiloop system with a traditional two-loop system, the authors of [9] found that the transfer efficiency could be better maintained by adopting the former approach. Through an ingenious coil structure design, the authors of [10] applied a cross coil at the receiving end, which improved the system’s ability to resist angular deviation. The authors of [11] used two orthogonal coils in a 3-D model to achieve orthogonal transmission, which is particularly good at resisting angular drift.

It has been reported that WPT systems with a hybrid metamaterial slab (combined with metamaterials of negative and zero permeability) can enhance efficiency and reduce the magnetic field in short distance systems [12,13]. A new structure using metamaterials was designed to improve the transfer efficiency, increasing efficiency by 28% [14]. In [15], a dual band metamaterial is proposed for use in portable devices, shielding the leakage magnetic field and focusing on the magnetic field. Metamaterials with a cavity can form a magnetic field in the field localization, providing power only to the intended localized zone [16]. Electromagnetic metamaterials have strong magnetic field control capabilities [17,18]. Many studies have shown that electromagnetic metamaterials can enhance the resonant coil coupling of WPTs and solve the electromagnetic leakage problem [19,20,21,22].

In this paper, we propose an innovative approach using electromagnetic metamaterials to improve the antioffset capability of WPT systems. Based on electromagnetic metamaterials design theory, we designed an electromagnetic metamaterial with a relative permeability of –1 at 3 MHz and an established four-coil WPT system. The experimental results showed that the WPT system with electromagnetic metamaterials was resistant to angular and lateral offset. The experimental results also agreed with the simulation results, which verified that the scheme for enhancing the stability of a WPT system proposed in this paper is feasible.

## 2. Theoretical Analysis and Simulation

In this paper, a four-coil structure model is selected. A schematic diagram of the equivalent circuit is shown in Figure 1. The system is composed of a driver coil, transmitter coil, receiver coil, and load coil; additionally, the transmitter and receiver coils adopt series capacitor (SS) topologies. In the picture, *R*_1_ and *R*_4_ are the equivalent resistances of the driver and load coils, respectively. *R_L_* is the charging load, and *L*_1_ and *L*_4_ are the coil self-inductances. *R*_2_, *R*_3_, *L*_2_ and *L*_3_ are the internal resistance and self-inductance of the transmitter and receiver coils, respectively. *C*_1_, *C*_2_, *C*_3_ and *C*_4_ are the compensation capacitors of each coil. *k*_12_ is the coupling coefficient between the driver coil and the transmitter coil, *k*_23_ is the coupling coefficient between the transmitter coil, and *k*_34_ is the coupling coefficient between the receiver coil and the load coil. $M_{ij}=k_{ij}\sqrt{L_{i}L_{j}}$ is the mutual inductance between two specific coils. *f* represents the system working frequency, so the angular frequency ω of the system can be expressed as 2π*f*. In medium distance transmission, *M*_13_, *M*_14_ and *M*_24_ are relatively weak, and can be ignored. Meanwhile, the portion of the virtual box in Figure 1 is the high frequency power supply, which is partially equivalent to the sinusoidal AC (alternating current) voltage source Vs and the internal resistance *R*s in series.

According to Kirchhoff’s voltage law (KVL), the four-coil system has the following relations,
(1)$$\left\{ \begin{array}{l} V_{s}=\left(R_{S}+R_{1}+j\omega L_{1}+\frac{1}{j\omega C_{1}}\right)I_{1}+j\omega M_{12}I_{2}\\ 0=\left(R_{2}+j\omega L_{2}+\frac{1}{j\omega C_{2}}\right)I_{2}+j\omega M_{12}I_{1}+j\omega M_{23}I_{3}\\ 0=\left(R_{3}+j\omega L_{3}+\frac{1}{j\omega C_{3}}\right)I_{3}+j\omega M_{23}I_{2}+j\omega M_{34}I_{4}\\ 0=\left(R_{4}+j\omega L_{4}+\frac{1}{j\omega C_{4}}+R_{L}\right)I_{4}+j\omega M_{34}I_{3} \end{array}\right.$$

In the expression (1), *I*_1_, *I*_2_, *I*_3,_ and *I*_4_ are the currents flowing through the four coils, respectively. It has been confirmed that the four coils are in a resonant state at the operating frequency of the system in a large number of studies on wireless transmission, which is a prerequisite for achieving the maximum transfer efficiency of the system. Simultaneously, when the positions of the transmitting and receiving coils are fixed (transfer distance is fixed), an optimal load value exists to maximize the system’s transfer efficiency. The optimal load value can be written as:(2)$$R_{43opt}=\frac{{\left(\omega_{0}k_{34}\sqrt{L_{3}L_{4}}\right)}^{2}}{R_{4}+R_{L}}=\sqrt{R_{3}^{2}+\frac{R_{3}}{R_{2}}(\omega_{0}k_{23}\sqrt{L_{2}L_{3}}{)}^{2}}$$

A high-frequency power module, i.e., a class E topological power amplifier, is adopted in this paper. In order to maximize the efficiency, the relationship between the electrical parameters of each discrete device in the power supply design process is shown in Table 1:

Combined with the structural characteristics of the four coils, it can be deduced that the load involved in the above power supply design process can be represented as:(3)$$R_{L}^{\prime }=R_{1}+\frac{{\left(\omega k_{12}\sqrt{L_{1}L_{2}}\right)}^{2}\left[{\left(\omega k_{34}\sqrt{L_{3}L_{4}}\right)}^{2}+R_{3}\left(R_{4}+R_{L}\right)\right]}{\left(R_{4}+R_{L}\right){\left(\omega k_{23}\sqrt{L_{2}L_{3}}\right)}^{2}+R_{2}\left[{\left(\omega k_{34}\sqrt{L_{3}L_{4}}\right)}^{2}+R_{3}\left(R_{4}+R_{L}\right)\right]}$$

It can be found that in the four-coil system with fixed transfer distance, there are two degrees of freedom, i.e., *k*_12_ and *k*_34_. By changing the distance between the transmitter coil and the driver coil, and adjusting the distance between the load coil and the receiver coil, *k*_12_ and *k*_34_ will change accordingly. Both *R*_43opt_ and $R_{L}^{\prime }$ will reach optimal values, so as to maximize the transfer efficiency of the four-coil system. By changing *k*_12_ and *k*_34_, the overall efficiency of the system changes, as shown in the Figure 2.

As can be seen in Figure 2, during the experiment, we first adjust *k*_34_ to make the equivalent load resistance of the wireless transfer system reach the optimal value. Then, by adjusting *k*_12_, the mapping parameters of the system meet the requirements of the power supply design. As such, the overall efficiency of the system is maximized. This is also the main reason for choosing a four-coil structure wireless transfer system in this paper. When the resonant coil is shifted, the matching condition of the system is broken, and the introduction of electromagnetic metamaterials and the regulation of *k*_34_ allow the system to meet the matching condition once again.

In this paper, a coil with spatial helical structure is used. When studying the misalignment, the coil had the following configurations: Figure 3a shows the position relationships of the coils without misalignment. Figure 3b shows the coils with lateral misalignment. Figure 3c shows the coils with angular offset. Figure 3d includes the coils with angular and lateral offset at the same time.

In the Figure 3, Δx is the lateral offset, α is the angular offset, $R_{t}$ and $R_{r}$ are the coil radius of the transmitter coil and the receiver coil respectively, and d is the distance between the coils.

The four cases shown in the above figures can be integrated into a set of calculation methods, where $d{\vec{l}}_{t}$ is the transmitter coil element, $d\vec{l_{r}}$ is the receiver coil element, and $R_{tr}$ is the distance between the wire elements.
(4)$$\begin{array}{l} d{\vec{l}}_{t}=R_{t}(-\sin\theta \vec{x}+\cos\theta \vec{y})d\theta \\ d{\vec{l}}_{r}=R_{r}(-\sin\phi \cos\alpha \vec{x}+\cos\phi \vec{y}+\sin\phi \sin\alpha \vec{z})d\phi \\ R_{tr}=\left|\begin{array}{l} (\Delta x+R_{r}\cos\phi \cos\alpha ,R_{r}\sin\phi ,d-R_{r}\cos\phi \sin\alpha )-\\ \left(R_{t}\cos\theta ,R_{t}\sin\theta ,0\right) \end{array}\right| \end{array}$$

Based on Newman’s formula, the mutual inductance between coils with misalignment can be expressed as follows.
(5)$$\begin{array}{l} M=\frac{\mu_{0}R_{t}R_{r}}{4\pi }∯\frac{d{\vec{l}}_{t}d{\vec{l}}_{r}}{R_{tr}}\\ =\frac{\mu_{0}}{4\pi }\iint \frac{\left(\sin\theta \sin\phi \cos\alpha +\cos\theta \cos\phi \right)}{{\left[\begin{array}{l} R_{t}^{2}+R_{r}^{2}+d^{2}+\Delta x^{2}+2\Delta xR_{r}\cos\phi \cos\alpha -2\Delta xR_{t}\cos\theta -\\ 2R_{t}R_{r}(\cos\theta \cos\phi \cos\alpha +\sin\theta \sin\phi )-2R_{R}d\cos\phi \sin\alpha \end{array}\right]}^{\frac{1}{2}}}d\phi d\theta \end{array}$$

Previous studies have shown that when the coil is a multiturn space helix, the coil can be equivalent to multiple single-turn coils in series. Since the mutual inductance between single-turn coils in various spatial relationships is solved in (5), the mutual inductance between the coils used in this paper can be calculated.

At the same time, based on the finite element simulation software, the mutual inductance of the space helical structure resonant coil is simulated under angular and lateral offsets. The simulation and calculation results are shown in Figure 4. It can be observed that the allowable error range and the mutual inductance of coils can be calculated; and the correctness of the calculation method of mutual inductance between the proposed resonant coils is verified in this paper. All analyses of mutual inductance in this paper will use this method.

Metamaterials are a new type of artificial electromagnetic functional materials. Since metamaterials are generally composed of arrays of periodic subwavelength structural units, their periodic structures interact with electromagnetic waves to produce complex electromagnetic responses; as such the materials usually exhibit supernormal electromagnetic characteristics to the outside world. Therefore, metamaterials can overcome the limitations of some natural laws, achieving supernormal material functions that would be impossible to achieve with a single natural material, such as negative refractive indexes and zero permeability.

Negative magnetic metamaterials (NMM) refer to a class of materials whose equivalent permeability is negative within a certain working frequency band. Previous studies have shown that both magnetic field focusing and evanescent wave amplification can be achieved by NMM. At the same time, the research presented in [14,15] described a method whereby NMM can be realized by adjusting the structural parameters of the material units. The structural parameters of the NMM unit in this paper were obtained through a large number of simulations. A schematic diagram is presented in Figure 5.

The magnetic field intensity distribution of a WPT system with and without NMM is shown in Figure 6. Figure 6a,b show the magnetic field intensity when angular offset occurs. Figure 6c,d show the magnetic field distribution when lateral misalignment occurs. It can be seen that the NMM can significantly enhance the magnetic field around the receiver coil end, regardless of the angular offset or the lateral misalignment, which proves that the NMM can enhance the misalignment tolerance of the WPT system.

## 3. Experimental Measurements and Results

The WPT system was designed in Figure 7. The system consists of a driver coil, transmitter coil (*T*x), receiver coil (*R*x), load coil, and electromagnetic metamaterials placed between *T*x and *R*x. The resonant frequency of the coils reaches 3 MHz by adjusting the capacitors connected in series with *T*x and *R*x. The load coil is connected to a pure resistance of 10 ohms. During the experiment, the system output power can be determined by simply measuring the voltage signal on the resistance; in turn, the overall transfer efficiency can also be determined. The *T*x and *R*x are helix coils with eight turns and a diameter of 50 cm. The driver and load coils are single-turn coils. The transmission distance *D_tr_* of the WPT system is fixed at 1m, which is the distance between the *T*x and *R*x.

The specific electrical parameters of the coils are shown in Table 2. In the table, *R*, *L*, and *F* respectively represent the coil resistance, coil inductance, and resonant frequency after capacitance matching.

The first experiment is to measure the efficiency of the WPT system without NMM. The distance between the *T*x and *R*x is fixed at 1 m. The experimental results are shown in Figure 8. The x-coordinate represents the distance between the load coil and the receiver coil, and the y-coordinate represents the system efficiency. It can be seen from Figure 8 that when the transfer distance is fixed, there is an optimal position between the load coil and the receiver coil, which can make the system reach maximum transfer efficiency. At this time, the transfer efficiency is 48.2%. The experimental results are consistent with the calculated results, which show the accuracy of the aforementioned theoretical model and the mutual inductance of the calculation method.

Further, we change the of the receiver coil’s position to keep its axis parallel to the transmission channel axis. The range of lateral misalignment is within one coil diameter, and the overall efficiency of the WPT system at each misalignment distance is obtained experimentally. The experimental results are shown in Figure 9. The horizontal coordinate represents the lateral misalignment; the measurement step length is 5 cm. As shown in the results, the system efficiency gradually decreases with the increase of misalignment distance. When the misalignment distance reaches 50 cm, the system efficiency drops to 30.74%, which is 18% lower than that without offset. Then, we put the NMM in the middle of the transmission channel. The experimental results show that when there is no lateral misalignment, the efficiency of the WPT system with NMM is improved, reaching 55.6%. Simultaneously, when the system is offset, NMM can reduce the impact of offset on the system. Even when the lateral offset reaches 50 cm, the system efficiency remains above 45.4%. The experimental results show that NMM can improve the transfer efficiency of the WPT system, and improve the stability of the system when lateral misalignment occurs. The NMM can enhance the lateral antioffset performance of the WPT system and maintain a high level of transfer efficiency.

Finally, we explore the angle offset of the system with and without NMM; the experimental results are shown in Figure 10. We keep the receiver coil central point unchanged and change the angle between the receiver coil axis and the transfer channel axis. The system is not loaded with NMM, and when the angle variation is small, the effect on the system is minimal, and system efficiency is maintained. When the angle exceeds 25 degrees, the system efficiency drops sharply. When the angle misalignment reaches 70 degrees, the system efficiency drops to 25%. Compared with the case without angle misalignment, the efficiency decreases by 23%, which is half of the original efficiency. After the introduction of EM, the declining trend of system efficiency slows down. When the angle deviation is 70 degrees, the system efficiency remains at 40.2%. The experimental results show that the NMM can enhance the resistance of the WPT system to angle misalignment and improve the stability of the system when such misalignment occurs.

## 4. Discussion and Expectation

NMM can mitigate the effects of angular deviation and lateral misalignment in WPT systems. Under the experimental conditions applied in this paper, the transfer efficiency of the system without loading NMM decreased by 18% with maximal lateral misalignment. In comparison, the efficiency of the system with NMM decreased by only 3%. At the maximum angular deviation, the efficiency of the system without NMM decreased by 23%, while the efficiency of the system with NMM decreased by only 8%. The experimental data show that NMM can enhance the lateral misalignment tolerance of the system by five times. The NMM was nearly twice as resistant to angular excursion. The ability to angular deviation increased almost two-fold. It was shown that the NMM can be used to maintain stability in wireless transmission systems. In future practical application scenarios, NMM could be regarded as a good choice for WPT technology to resist misalignment.

Currently, work in this area is still in progress. The design of NMM mainly relies on a large number of simulations and the specified apparatus, which significantly limits the usage scenarios and extends the design cycle of the system. However, the preliminary results are encouraging, and for the specified WPT system, we could design a corresponding NMM that could maintain the transfer efficiency of the system at a high level, even when the resonant coil is offset.

In this paper, we only presented an efficiency comparison before and after the NMM was loaded in the system, and only when there was angular offset or lateral misalignment. In the future, we could further explore more general situations in which both lateral and angular offset occurs. At the same time, whether the NMM design method can be further simplified and whether the NMM design scheme, independent of the concrete object, can be found will be the direction of future research.

## Figures and Tables

**Figure 1 materials-13-05695-f001:**
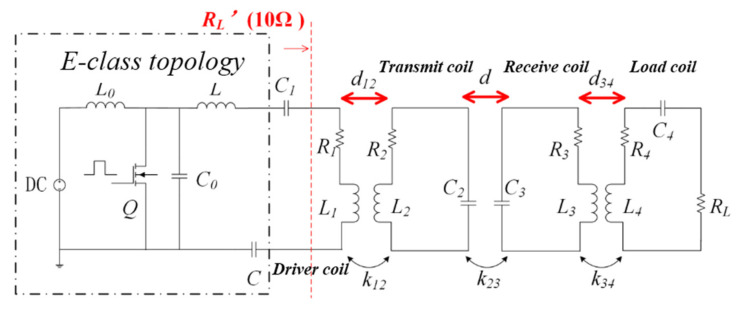
Wireless power transfer circuit topology.

**Figure 2 materials-13-05695-f002:**
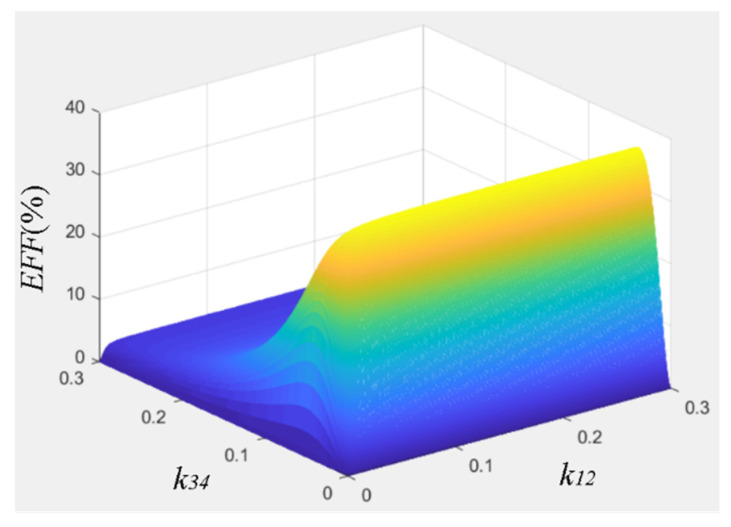
Transfer efficiency of the overall system.

**Figure 3 materials-13-05695-f003:**
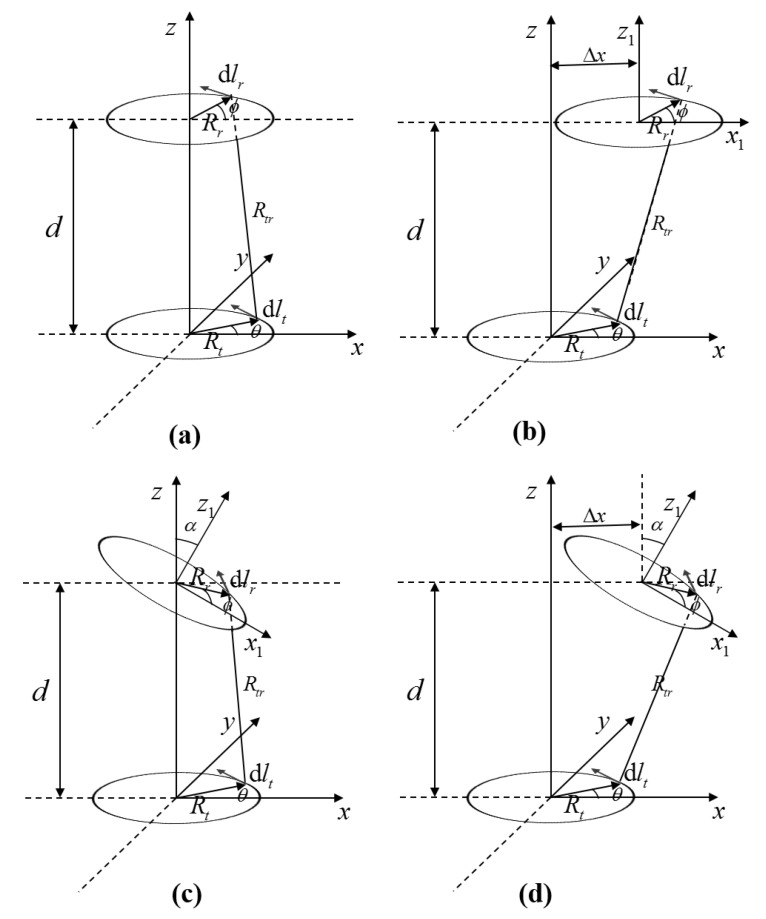
Schematic diagram of coil relative position: (**a**) no misalignment, (**b**) only lateral misalignment, (**c**) only angular offset, (**d**) a general case including both angular and lateral offset.

**Figure 4 materials-13-05695-f004:**
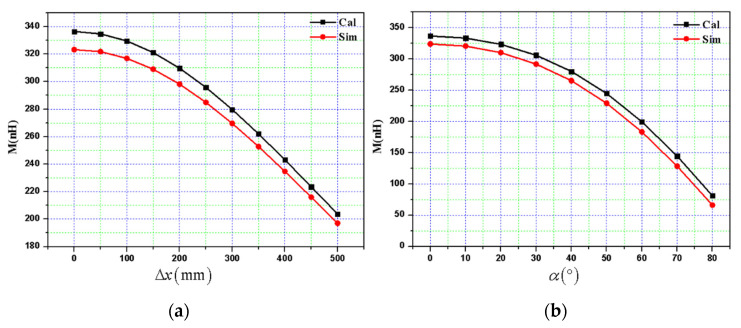
Calculation and simulation results of mutual inductance between resonant coils: (**a**) the relationship between mutual inductance and lateral offset, and (**b**) mutual inductance with the variation of angle.

**Figure 5 materials-13-05695-f005:**
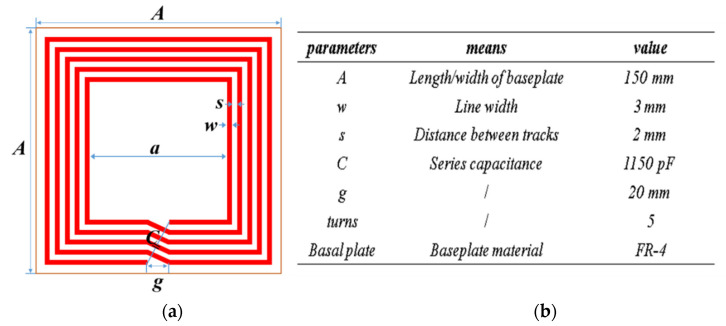
Schematic diagram and parameters of NMM structural units: (**a**) size structure diagram of the material, and (**b**) the meaning and actual size of the NMM in (**a**).

**Figure 6 materials-13-05695-f006:**
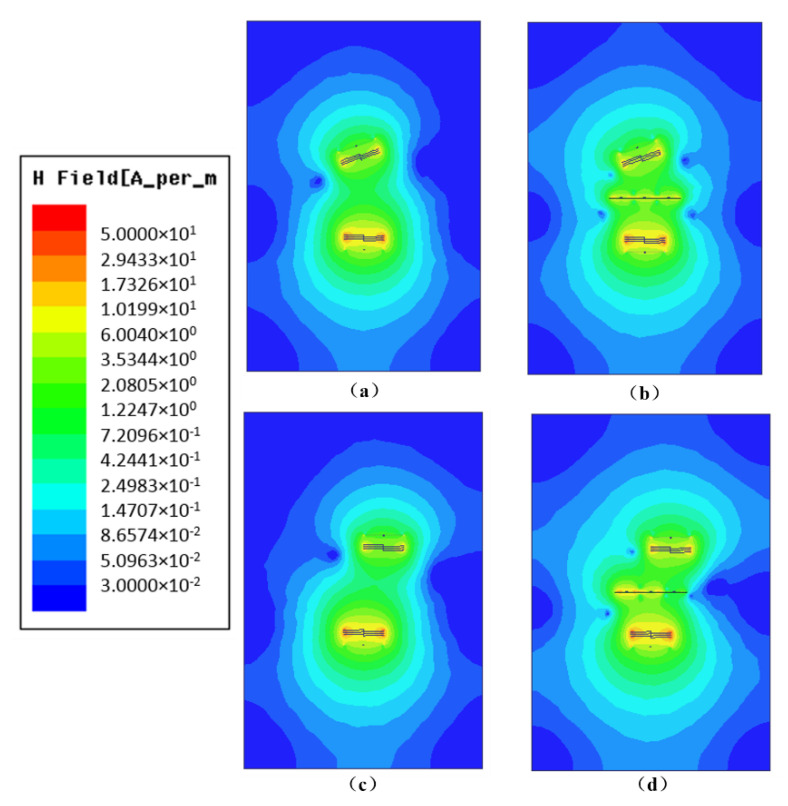
Comparison of magnetic field intensity before and after introducing NMM: (**a**) WPT system with angular offset, (**b**) the system in (**a**) and the system including NMM, (**c**) WPT system with lateral misalignment, and (**d**) adding NMM to (**c**).

**Figure 7 materials-13-05695-f007:**
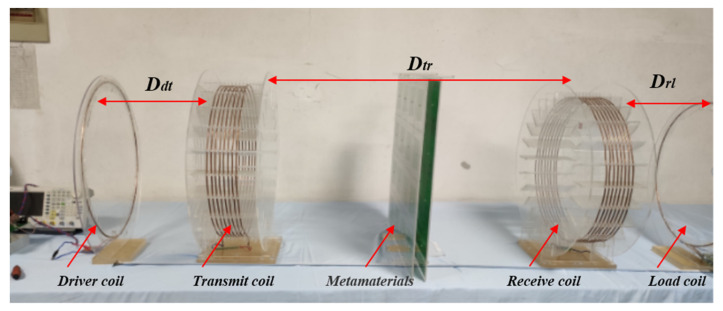
Physical picture of the experimental apparatus.

**Figure 8 materials-13-05695-f008:**
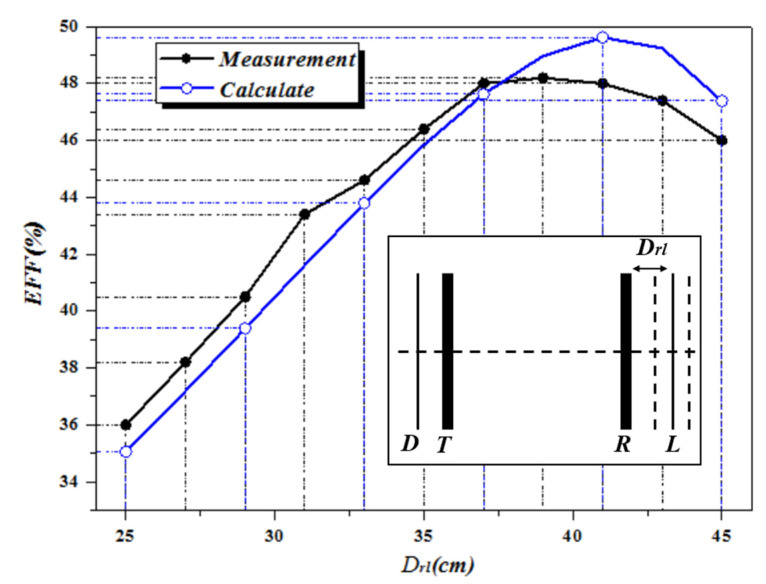
Calculation and simulation efficiency of WPT system without NMM.

**Figure 9 materials-13-05695-f009:**
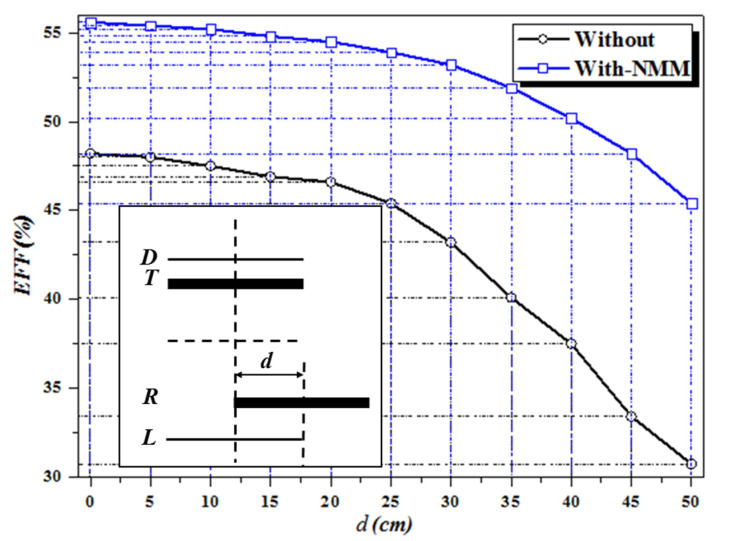
Comparison of system efficiency before and after NMM loading during lateral misalignment.

**Figure 10 materials-13-05695-f010:**
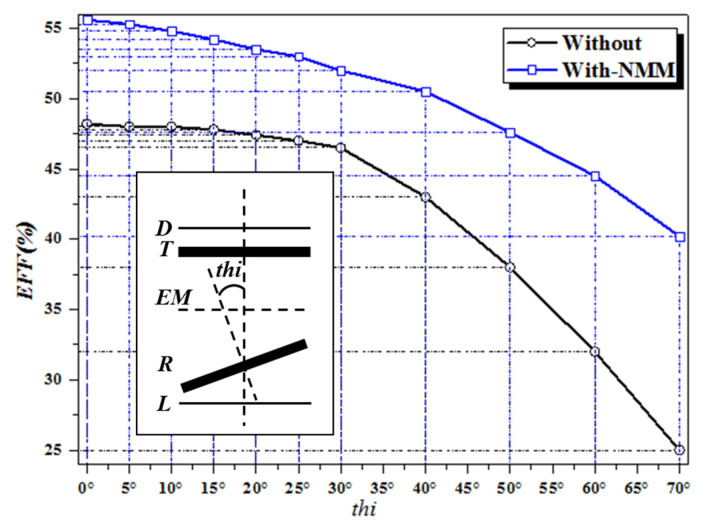
Comparison of system efficiency before and after NMM loading with angular offset.

**Table 1 materials-13-05695-t001:** Values range of parameters of the class E topological power amplifier.

	L_0_	C_0_	L	C
Range	$\geq \frac{1.25\left(4+\pi^{2}\right)\pi R_{L}^{\prime }}{\omega }$	$\frac{8}{\left(4+\pi^{2}\right)\pi \omega R_{L}^{\prime }}$	$\frac{Qr\times R_{L}^{\prime }}{\omega }$	$\frac{1}{\omega^{2}L-\omega R_{L}^{\prime }\tan\varphi }\left(\varphi =49.052^{\circ }\right)$

**Table 2 materials-13-05695-t002:** Electrical parameters of the coils.

	Driver Coil	Transmitter Coil	Receiver Coil	Load Coil
*R* (ohms)	0.17	2.38	2.41	0.18
*L* (uH)	1.56	54.61	54.82	1.59
*F* (MHz)	3	3	3	3

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
