# Peer review of "Enhancing the Stability of Medium Range and Misalignment Wireless Power Transfer System by Negative Magnetic Metamaterials"

_materials, 2020, doi:10.3390/ma13245695_

Round 1

Reviewer 1 Report

In the paper is devoted to the interesting investigation of the increasing of stability of wireless power transfer in the situation when misalignment of resonant coils in the systems tales place. The authors offer to use magnetic metamaterials for solving this problem.  In the paper detail experimental and theoretical study, including theoretical analysis and simulation including consideration of resistances of driver, load coil, charging load, inductances, internal resistance and self-inductance of the transmit and received coil, was provided. The experiments were carried out at the frequency 3 MHz. The results show that the system transfer efficiency can be stabilized for than 40%, if the variation of angle will be about 70 degrees. This is very interesting for practical application.  

The paper is comprehensive and good described.

The disadvantages of the paper are:

  1. Figure captions are not enough detail described.
  2. Also the description of experimental set up is not clear.

After the correction the paper can be published.  

Reviewer 2 Report

In this work, the authors discuss the effect which the use of the specific metamaterial has on the efficiency of the wireless power transfer system subjected amongst other to the misalignment. To this aim, the authors utilise several different approaches including the experiment which allows to validate the results obtained in this work. Overall, I think that this work is quite interesting and can be considered for publication once the following issues are addressed:

1) The authors may consider adding a graphical representation of the considered metamaterial to Fig. 1

2) It would be beneficial if authors were to amend Fig.6 so that it would be clear that panels a) and b) as well as c) and d) should be compared with each other and analysed together.

3) I think that the current title might be misleading for the readers. In their work, the authors write that their approach is based on the use electromagnetic metamaterials with the objective being to improve the anti-offset capability of WPT systems („In this paper, an innovative approach is proposed to use electromagnetic metamaterials to improve the anti-offset capability of WPT systems”). On the other hand, the title can be interpreted as if the misalignment was the main tool used in order to improve the efficiency of the WPT system.

4) There are several places in the manuscript where the quality of writing must be improved. For example, the authors wrote „… this paper designed electromagnetic metamaterials …” instead of referring to the authors being responsible for the design. In fact, there are quite a few errors of a different nature.

5) The literature review used in this paper is not particularly extensive. I think that the authors may expand it for example by discussing relevant achievements in the field of metamaterials

Reviewer 3 Report

In my opinion this paper can be interesting to readers of Materials journal. The paper is rather clearly presented. The paper contains 10 figures, 2 tables and 5 formulas – figures are legible and good quality. Authors added Highlights to submitted paper.

English of the paper is rather good and meet the requirement of the journal – in my opinion the language of the paper should be improved. I am asking for corrections by a native speaker.

I also find some mistakes for example:

  • Authors should correct Abstract – please describe the scientificity of the test results obtained.
  • In the whole paper, you write the values in percent as for example 48.2% (for example: line 212) – you should write and value with unit with spaces (48.2 %).
  • In Figures 2, 4, 8, 9 and 10 – units should be placed after commas – not in parentheses.
  • In the caption to Figure 3, all the component figures (a) ..., (b) ..., (c) ... and (d) ... should be indicated in the caption.
  • In the caption to Figure 4, all the component figures (a) ... and (b) ... should be indicated in the caption. Figures themselves should also be signed.
  • In the caption to Figure 6, all the component figures (a) ..., (b) ..., (c) ... and (d) ... should be indicated in the caption.
  • All dimensions in Figure 5 should be increased – too small size of markings – dimensions illegible.
  • Please increase the markings in the legend at Figure 6 – the size of the markings is too small – the size is illegible.
  • Chapters Theoretical analysis and simulation and 3. Experimental measurements and results  should be subsections in the chapter called "Discussion".
  • The final chapter in the article is "Discussion and expectation". There is no summary chapter – "Conclusions". In this chapter there are no summary of all significant research results obtained by the Authors and written in the Results
  • There are no references to literature next to formulas (1-5). Please indicate source literature.
  • Amount of references is also sufficient but some papers cited in the references (1 from all 19) are older then 10 years and 9 from all 19 are older then 5 years – I can state that there are enough new papers in the list of references.
  • Minimum 14 papers from all 19 are wrote by authors from Asia – China, Japan, Pakistan and others. I propose to add some new (from the last 5 years) publications. Authors should include several modern papers (also from Europe and America).
  • I the list of references I found 1 (from all 19) papers of the Authors of reviewed paper. Are these the first studies related to this subject for the Authors?

The manuscript can be accepted for publication in Materials journal after MAJOR corrections.

Round 2

Reviewer 3 Report

Authors have properly addressed the concerns from the referee. Most my remarks have been included in the revised document. They prepared answers in 3 points. Below you will find my comments on the attached answers.

Point 1 and Response 1:

Thank you for making corrections in line with my comments. However, the text of the article still contains incorrect entries of numerical values and their units (for example: line 58, 208, 212, 241, 242, 247 and others).

Point 2 and Response 2:

Thanks for the clarification. The answer is satisfactory

Point 3 and Response 3:

Authors have changed the list of references – they added few global and modern references. Changes are satisfactory. Moreover, English of the paper is rather good – in my opinion the language of the paper should be improved. I am asking for corrections by a native speaker.

Finally, the manuscript can be accepted for publication in Materials journal after preparing mentioned corrections – MINOR corrections.

Author Response

Dear reviewer,

     Thank you very much for your suggestion. In the latest version of the manuscript, according to your suggestion 1, some missing points have been modified. Meanwhile, thank you for your affirmation. Of course, we have also made some adjustments to the overall expression of the manuscript, hoping that it can meet your requirements.